# Rare Earth Group Separation after Extraction Using Sodium Diethyldithiocarbamate/Polyvinyl Chloride from Lamprophyre Dykes Leachate

**DOI:** 10.3390/ma15031211

**Published:** 2022-02-05

**Authors:** Eman M. Allam, Taysser A. Lashen, Saeyda A. Abou El-Enein, Mohamed A. Hassanin, Ahmed K. Sakr, Mohamed F. Cheira, Aljawhara Almuqrin, Mohamed Y. Hanfi, M. I. Sayyed

**Affiliations:** 1Nuclear Materials Authority, El Maadi, Cairo 11381, Egypt; ah841873@ucf.edu (T.A.L.); mmsh236@yahoo.com (M.A.H.); akhchemist@gmail.com (A.K.S.); mokhamed.khanfi@urfu.ru (M.Y.H.); 2Department of Chemistry, Faculty of Science, Menoufia University, Shebin El-Kom 32511, Egypt; dr.saeyda_elenein@yahoo.com; 3Department of Physics, College of Science, Princess Nourah Bint Abdulrahman University, P.O. Box 84428, Riyadh 11671, Saudi Arabia; ahalmoqren@pnu.edu.sa; 4Institute of Physics and Technology, Ural Federal University, St. Mira, 19, 620002 Yekaterinburg, Russia; 5Department of Nuclear Medicine Research, Institute for Research and Medical Consultations (IRMC), Imam Abdulrahman Bin Faisal University (IAU), P.O. Box 1982, Dammam 31441, Saudi Arabia; mabualssayed@ut.edu.sa; 6Department of Physics, Faculty of Science, Isra University, Amman 11622, Jordan

**Keywords:** rare earth ions, sodium diethyldithiocarbamate, polyvinyl chloride, sorption, desorption, RE concentrate

## Abstract

This study presents the first application of sodium diethyldithiocarbamate/polyvinyl chloride (DdTC/PVC) as a novel adsorbent for rare earth element (REE) sorption from leach liquors. DdTC/PVC has higher adsorption properties than other sorbents, the synthesis of DdTC/PVC is more accessible than other resins, and it is considered a more affordable sorbent. The three-liquid-phase extraction technique (TLPE) was applied to separate REEs into light, middle, and heavy rare earth elements as groups. The TLPE is an excellent achievable technique in the separation of REEs. DdTC/PVC was prepared as a sorbent to sorb rare-earth ions in chloride solution. It was described by XRD, SEM, TGA, and FTIR. The factors pH, initial rare-earth ion concentration, contact time, and DdTC/PVC dose were also analyzed. The ideal pH was 5.5, and the ideal equilibration time was found to be 45 min. The rare-earth ion uptake on DdTC/PVC was 156.2 mg/g. The rare-earth ion sorption on DdTC/PVC was fitted to Langmuir and pseudo-2nd-order models. The rare-earth ions’ thermodynamic adsorption was spontaneous and exothermic. In addition, rare-earth ion desorption from the loaded DdTC/PVC was scrutinized using 1 M HCl, 45 min time of contact, and a 1:60 S:L phase ratio. The obtained rare earth oxalate concentrate was utilized after dissolving it in HCl to extract and separate the RE ions into three groups—light (La, Ce, Nd, and Sm), middle (Gd, Ho, and Er), and heavy (Yb, Lu, and Y)—via three-liquid-phase extraction (TLPE). This technique is simple and suitable for extracting REEs.

## 1. Introduction

Rare-earth ions are employed in a sweeping range of applications and techniques. A wide range of applications depend upon their catalytic, chemical, electrical, optical, and magnetic properties. REEs are widely utilized in classic sectors, including petroleum, metallurgy, agriculture, and textiles. They also have substantial and unique uses in numerous industries such as compact lights, wind turbines, hybrid cars, mobile phones, screen televisions, defense technologies, and disc drives [1,2,3,4,5,6,7,8,9]. REEs are formed in the earth’s crust during the process of fractional crystallization in molten rock; they are concentrated in residual fluid that crystallizes as pegmatite dikes, and so great number of their minerals occur chiefly in pegmatite dikes associated with igneous rocks and in deposits derived from the weathering of pegmatite [10].

On the other hand, chemical alteration of the original igneous minerals has produced rare earth minerals such as bastnasite [11]. In addition, other rare earth resources involve euxenite, gadolinite, apatite, pyrochlore, and non-rare earth mineral resources, especially igneous rocks containing uranium and thorium [12]. Rare earth elements (REEs) are recovered from ore materials, requiring some hydrometallurgical processes. Amazing efforts have been concentrated upon separating REEs from new materials. Impregnated resins originated by altering the solid support for metal separation from complex matrices. Two processes have embraced to organize solid sets. One is based on the physical modification of a suitable solvent at solid support. The second involves tying the chelating complex to support materials. For the extraction and separation of particular REEs, various solid-extraction techniques have been conceived using distinct styles of solid supports, such as activated carbon, silica, clay, titanium oxide, polymeric resins, Dowex 50 × 8, Dowex X1, and naphthalene [13].

A resin was proposed for the extraction of lanthanides from acidic wastes. Nonionic amberlite XAD-16 polymeric was impregnated with undiluted TBP to extract Ce^4+^ from nitrate solution at room temperature. The cerium loading capacity of the impregnated resin reached 95.6% of the calculated theoretical capacity (173 mg/g) under the conditions of a solution to resin ratio of 10.0 and a contact time of 5.0 min [14]. A mono aza dibenzo crown ether/amberlite XAD-4 was applied to extract Sm^3+^, La^3+^, and Nd^3+^ from a synthetic solution at pH 4.5 [15]. Lanthanum and cerium were separated using calix [4]arene-o-vanillinsemicarbazone immobilized on a polymeric matrix—a Merrifield peptide resin at pH 5–8 [16]. Dowex 50X8 resin was used for REE separation from chloride solution at pH 6 [17]. La^3+^ and Ce^3+^ species were extracted using ethylhexyl phosphonic acid mono-2-ethylhexyl ester covered by polyvinyl alcohol that was crosslinked by divinyl sulfone or glutaraldehyde [18].

Recently, foam flotation methods have become a promising technique of recovering RE elements from leachates of primary and secondary resources. Foam flotation can recover La^3+^, Ce^3+^, Gd^3+^, and Yb^3+^ ions individually, as a group, and as a group with gangue ions (Al^3+^, Zn^2+^, Ca^2+^, and Mg^2+^). The anionic surfactants sodium dodecyl sulfate (SDS) at basic pH and mono-rhamnolipid at pH 9 were used to separate REEs in the presence of gangue cations with a surfactant; the total cation (excluding Na+) ratio was as low as 1:13 [19]. Rare earth minerals enriched pre-concentrate (after lab-scale gravity and magnetic separation steps) were separated by hydroxamic acid flotation [20]. Copper oxide sulfidization flotation was applied after the modification of cuprite by hydrogen peroxide (as an oxidant to improve its consequent sulfidization) [21]. Moreover, the direct sulfidization of cuprite is ineffective because cuprite is a copper-oxide mineral with strong surface hydrophilicity. The flotation recovery of pre-oxidized cuprite was nearly 25% higher than that of direct sulfidization flotation, which indicates that the cuprite surface activity was enhanced after pre-oxidation by the transformation of Cu^+^ species (weak affinity with sulfur ions) to Cu^2+^ species (strong affinity with sulfur ions) [22].

The heavy rare-earth elements were adsorbed by hexyl-ethyl-octyl-isopropylphosphonic acid anchored resin from 0.2 M hydrochloric acid [23]. Bis(2-ethylhexyl)phosphoric acid/polyethersulfone polymer was prepared and tested for its suitability for separating rare-earth ions from chloride media at 0.5 M containing La^3+^, Sm^3+^, and Y^3+^ [24]. The chromatographic separation of the impurities Ce^3+^, Pr^3+^, Nd^3+^, Sm^3+^, Zn^2+^, Al^3+^, Ca^2+^, and Fe^3+^ from La^3+^ was examined using bis(2-ethylhexyl)phosphoric acid impregnated Amberlite™ XAD-7 HP from HCl solution [25]. La_2_O_3_ was extracted from monazite that was treated in three steps; (a) lanthanum hydroxide extraction by using NaOH, (b) digestion with HNO_3_, (c) precipitation with NH_4_OH, and calcination to La_2_O_3_ [26]. Tributyl phosphate (TBP) was used to extract lanthanum, cerium, yttrium, and neodymium from an aqueous solution produced by nitric acid leaching of apatite concentrate by solvent extraction at 3.65 mol/L TBP in kerosene, 5 min, 25 °C, and a 1:1 organic/aqueous ratio [27].

TLPE was utilized as the group extraction of multi-component approaches owing to its unique extraction of three-liquid-phases via several physiochemical effects [28,29,30]. A straightforward TLPE strategy was applied to attain group separation of the light, middle, and heavy rare-earth ions by liquid–liquid–liquid three-phase systems. The heavy rare-earth ion Yb^3+^ was selectively extracted in the top organic phase. In contrast, the middle rare-earth ion Eu^3+^ and the light rare-earth ion La^3+^ can be enriched, respectively, in the PEG-rich middle phase and the (NH_4_)_2_SO_4_-rich bottom aqueous phase of the Cyanex 272/PEG/(NH_4_)_2_SO_4_–H_2_O three-liquid-phase system. Moreover, the three-liquid-phase employed the recovery of REEs from liquid liquor [31,32].

Herein, hydrochloric acid was used to liquefy rare earth ions from Abu Rashied Lamprophyre dykes in the southeastern desert of Egypt. The prepared sodium diethyldithiocarbamate trihydrate/polyvinyl chloride (DdTC/PVC) was utilized to adsorb REEs from hydrochloride solution. Various influences of aqueous pH, (NH_4_)_2_SO_4_ concentrations, complexing agents, polymers, and added amounts on the three-liquid-phase separation of light, middle, and heavy rare-earth ions were examined. Furthermore, a possible partitioning mechanism for three rare-earth ions was analyzed. The optimum sorption conditions and three-liquid phase of REEs were also found. A three-liquid extraction phase was explored to isolate light, middle, and heavy rare-earth as different groups.

## 2. Materials and Methods

### 2.1. Materials and Instrumentations

All materials utilized in all different sections were of analytical grade and they were used without further purification. The polyvinyl chloride was impregnated with sodium diethyldithiocarbamate (DdTC) in the laboratory. REEs (1000 mg/L) were obtained from Loba Chemie, Mumbai, India. Diethylenetriamine penta-acetic acid (DTPA), ammonium hydroxide, hydrochloric acid, ammonium sulfate, and phosphoric acid were obtained from BDH, Poole, UK. Cyanex 272 and three kinds of polyethylene glycols (600, 2000, 6000 molecular weight of PEG) were obtained from Sigma-Aldrich, St. Louis, MO, USA.

A double beam spectrophotometer (Unicam, Cambridge, UK) with 1 cm quartz cells covering the UV-visible range of 200–1100 nm was employed for the determination of REEs and other major oxides in all aqueous samples. In addition, Na^+^ and K^+^ were determined by flame photometer (Model 410, Sherwood, Cambridge, UK). Atomic absorption and inductively coupled plasma-optical emission spectrometer (Agilent, Santa Clara, CA, USA) techniques were utilized to analyze the trace metal ions. The X-ray diffraction approach (XRD) (Malvern Panalytical, Almelo, The Netherlands) was also used to categorize the materials’ constituents. The material groups were explored via a Fourier transform infrared (FTIR) spectrophotometer (Shimadzu, Kyoto, Japan). The specific surface area and total pore volume of prepared materials were assessed using nitrogen sorption at 77 K (USA).

### 2.2. Preparation of Sodium Diethyldithiocarbamate Trihydrate/Polyvinyl Chloride

The dry technique was used to prepare sodium diethyldithiocarbamate trihydrate/polyvinyl chloride (DdTC/PVC). Five grams of polyvinyl chloride were suspended in 70 mL double-distilled H_2_O. Then, it was mixed well with 0.5 g of sodium diethyl-dithiocarbamate, which was dissolved in 70 mL ethyl alcohol. The mixture was stirred under reflux for 3 h at 22 °C until complete homogenization and then left to evaporate on a hotplate. The modified polyvinyl chloride was dried at 65 °C.

### 2.3. REE Sorption Studies

In the current study, various experiments were conducted to determine the relevant parameters affecting the sorption of REEs from either the synthetic solution or leach liquor on sodium diethyl-dithiocarbamate anchored polyvinyl chloride (DdTC/PVC). The studied parameters were aqueous pH, initial RE ion concentration, sorbent dose, contact time, and temperature.

A series of investigations was undertaken using different concentrations of the REE synthetic solution with a constant volume at 200 rpm to determine the optimum parameters. The influence of pH upon REE sorption was investigated with pH 1 to 7. The influences of sorbent dosage (10 to 90 mg) and agitation time (5 to 120 min) were also studied. The initial RE ion concentrations were studied in the 25–400 mg/L range. The effect of temperature was analyzed under various temperatures. These parameters were applied to the DdTC/PVC adsorbent by batch strategy. All the studies were executed in duplicate. The sorption power *q_e_* (mg/g), sorption efficiency (*E*, %), and distribution coefficient (*K_d_*) were summed by Equations (1)–(3) [33,34,35].
(1)$$q_{e}=\left(C_{0}-C_{e}\right)\times \frac{V}{m}$$
(2)$$E\;\left(\%\right)=\left(\frac{C_{0}-C_{e}}{C_{0}}\right)\times 100$$
(3)$$K_{d}=\left(\frac{C_{0}-C_{e}}{C_{0}}\right)\times \frac{v}{m}$$

*C_e_* and *C*_0_ (mg/L) denote the equilibrium and initial REE concentrations, *V* (L) and *v* (mL) are solution volumes, and *m* (g) is a sorbent weight.

### 2.4. Desorption Studies

The rare earth-loaded DdTC/PVC was used to study desorption characteristics. The desorption processes were performed with 0.5 g of REEs/DdTC/PVC treated with 25 mL of adaptable concentrations of hydrochloric, nitric, and sulfuric acids ranging from 0.1 to 3 M for 60 min contact time and then filtered. The eluting RE ions were determined. The effect of desorption parameters like the desorbent type, the concentration of eluent, contact time, and temperature were investigated. After treating the studied adsorbent with the desorbing agent, it was carefully washed through double-distilled water to be eligible for recycling.

### 2.5. Precipitation of REEs

After the REEs sorption-desorption procedure, the eluting REEs were precipitated by an excess oxalic acid (20.0%) at pH 1–2. The latter processes are continually used in industrial operations due to the processes’ simplicity and effective recovery. The mixture was left for 24 h to complete precipitation, and the precipitate was filtered.

### 2.6. Group Separation of REEs by Three-Liquid-Phase Technique

TLPE is a unique technique for convoluted multi-component strategy group extraction due to its superior separation. A sufficiently conceived three-liquid-phase that is an achievable method of enhancing other ingredients to three separate liquid phases. Hence, a one-stage separation of multi-components is achievable. Three-liquid extraction phases were explored to separate light, middle, and heavy rare-earth into groups from REE leachate. TLPE is composed of three phases: Cyanex 272 (top organic phase), polyethylene glycol (middle polymer phase), and NH_4)2_SO_4_ (bottom salt phase). The influence of pH, phase concentration, and complexing intermediates on REE partitioning were studied.

## 3. Results and Discussion

### 3.1. Characterization

#### 3.1.1. XRD Analysis

Figure 1 displays the X-ray diffraction configuration for the PVC polymer, DdTC/PVC, and REEs/DdTC/PVC. It is well known that PVC polymer exhibits a semi-crystalline structure with an amorphous nature due to the presence of peaks at the 2*θ* angles of 12.0°, 16.0°, 25.0°, and 39.5°. The broad peak at 2*θ* = 25.0° indicates the amorphous nature of PVC [36]. The three peaks at 2θ = 12.0°, 16.0°, 25.0° merged to one peak at 2*θ* = 22.0° for DdTC/PVC; moreover, the peak at 2*θ* = 39.5° disappeared for DdTC/PVC. Therefore, it was revealed that DdTC was impregnated into PVC to form DdTC/PVC due to the surface electrostatic interaction. The XRD of REEs/DdTC/PVC was also obtained, which detected some new peaks at 2*θ* = 10.0°, 15.0°, 19.0° due to rare-earth ion adsorption onto DdTC/PVC by the chemical interaction procedure.

#### 3.1.2. SEM/EDX Analysis

The SEM was used to scrutinize the modification of surface and physical realizations of DdTC/PVC before and after REE adsorption. Figure 2a illustrates that the surface of DdTC/PVC was characterized by minor aggregations with regular and slightly globular structures. The loaded REEs in the DdTC/PVC photographs in Figure 2b demonstrate that the particles were agglomerated, and the REEs formed bright spots on the DdTC/PVC adsorbent. From the results, it is evident that the surface morphology changes in the REEs/DdTC/PVC confirmed that the adsorption of REEs took place.

EDX of DdTC/PVC before and after REE adsorption is given in Figure 2c,d. In contrast, Figure 2c demonstrates that the N, C, S, and Cl peaks were accessible in the spectrum of DdTC/PVC. These results confirmed that the DdTC/PVC was established and formed. Furthermore, Figure 2d, which shows RE ion adsorption on DdTC/PVC, shows that distinct peaks of some RE ions (Y, Ce, La, Nd, Pr, Sm) appeared. It can be concluded that the REE peaks were perceived, confirming REE adsorption on DdTC/PVC.

#### 3.1.3. BET Surface Analysis

Figure 3a,b display the nitrogen adsorption-desorption isotherm curves of the studied materials. The isotherms were clearly changed after DdTC/PVC was adsorbed with REEs. The data in Table 1 reveal that the specific surface area (S_BET_), pore-volume, and pore size of DdTC/PVC were 90.14 m^2^/g, 0.087 cc/g, and 1.97 nm, respectively. From the data, it can be seen that the specific surface area, pore-volume, and pore size of the DdTC/PVC adsorbent decreased after rare-earth ions adsorption because the active sites were blocked with rare-earth ions. The obtained data shows that REEs were strongly adsorbed on the DdTC/PVC because they had more active sites than the PVC polymer.

#### 3.1.4. FTIR Analysis

The DdTC/PVC and REEs/DdTC/PVC materials were characterized by FTIR spectroscopy, with results shown in Figure 3c,d. The FTIR of sodium diethyldithiocarbamate trihydrate/polyvinyl chloride showed a broad peak at 3357 cm^−1^ due to the hydrated water’s O-H group (Figure 3c). Furthermore, the peaks at 2965–2846 cm^−1^ were assigned to the –CH_2_ and –CH– groups. The distinctive peak at 1593 cm^−1^ was related to the C=C of PVC. The peak at 1425 cm^−1^ belonged to C=S, while peaks at 1194 cm^−1^ and 1058 cm^−1^ fitted with the stretching C-N group [37]. Moreover, the peaks at 752 and 690 cm^−1^ were related to the C=S and C-S groups [38].

The obtained results presented in Figure 3d show the main differences between the above data before and after the adsorption of rare-earth ions on the DdTC/PVC. After REE adsorption, the vibration peaks of the DdTC/PVC adsorbent condensed and shifted to 5–10 cm^−1^, which may be due to the REEs adsorbing on the DdTC/PVC. Moreover, new peaks appeared around 874–472 cm^−1^ due to the REE adsorption. Furthermore, the peaks of C=S appeared at 747 and 700 cm^−1^ in the spectrum of REEs/DdTC/PVC [39,40]. This means that the rare-earth ions were adsorbed and reacted with the DdTC/PVC. Consequently, it is understood that DdTC/PVC appeared more friendly to REE adsorption.

#### 3.1.5. Thermal Analysis

Thermogravimetric examination (TGA) was used to study the thermal stability at a 10 °C/min temperature rate between 25–900 °C under an N_2_ environment. Figure 3e,f present the TGA thermograms of DdTC/PVC and REEs/DdTC/PVC with three stages of weight loss [39,40]. The first stage of weight loss at 100–115 °C was due to the loss of H_2_O. The second weight-loss stage occurred at 255–345 °C due to the elimination of HCl, NH_3_, and sulfur from the DdTC/PVC adsorbent. The third stage appeared at 350–650 °C due to the thermal degradation of the carbon chains, which produced flammable volatiles. The remained residue of DdTC/PVC was 4.33%, while the residue of REEs/DdTC/PVC was 10.96%. Hence, the REEs/DdTC/PVC residue was higher than the DdTC/PVC residue due to REE adsorption and presence on the DdTC/PVC surface.

### 3.2. REE Adsorption

Batch techniques were performed to explore the REE uptake upon DdTC/PVC from the REE aqueous solution. They were conducted by combining different amounts of DdTC/PVC adsorbent and 50 mL of REE solution. Several experiments were performed to optimize the pH, contact time, DdTC/PVC dose, initial RE ion concentration, and temperature, and to determine REE adsorption uptake.

#### 3.2.1. Effect of pH

The influence of pH upon REE adsorption efficiency from the solution was analyzed, and the results are exhibited in Figure 4a. Different experiments were executed in different pH ranges from 1 to 7. At the same time, the other conditions were kept constant, with 50 mL solution containing 200 mg/L REE concentration and a 50 mg adsorbent dose of DdTC/PVC, with a 30 min contact time. The data obtained established that the adsorption of REEs was gradually increased from 24.63 to 58.2% by increasing the pH from 1.0 to 5.5. 

The uptake of REEs was reduced in a higher acidic medium due to the existence of hydrogen ions that have a significant ability to adsorbed on the adsorbent active sites. Thus, hydrogen ions competed with the RE cations during the adsorption processes. The concentration of hydrogen ions decreased with increasing pH. The adsorption efficiency increased to maximum adsorption around pH 5.5. At pH > 5.5, lower REE adsorption values were observed because some RE ions were precipitated as hydroxide forms. Consequently, pH 5.5 was the optimum pH for running the subsequent experiments on REE adsorption.

#### 3.2.2. Effect of Adsorbent Dose

A series of experiments were executed using DdTC/PVC in a dose range of 10–90 mg at pH 5.5 and 50 mL of 200 mg/L RE ions. The acquired results are illustrated in Figure 4b, which reveal that the RE ion uptake on DdTC/PVC decreased from 156.6 to 89.1 mg/g at the 90 mg dose. At a higher dose of DdTC/PVC, more ion exchange sites were available for ion exchange procedures due to the increment of the surface area. Hence, a 50 mg DdTC/PVC dose was chosen as the sorbent dose for the subsequent trials.

#### 3.2.3. Contact Time and Kinetics

The role of sorption time on the REE sorption on DdTC/PVC was analyzed in the range of 5–120 min; the adsorption parameters were constant at pH 5.5, 50 mg DdTC/PVC adsorbent dose, and 50 mL REEs (200 mg/L) at room temperature. As seen in Figure 5a, the REE sorption uptake was improved by increasing the contact time until it reached equilibrium at 60 min, at which point REE uptake was ≈156.1 mg/g. The uptake remained constant up to 120 min.

The sorption reaction mechanism and kinetics were measured to fit the investigational data. Different kinetics models were executed on experimental data to analyze REE sorption kinetics upon DdTC/PVC.

Pseudo-first-order and pseudo-second-order models were applied for the reliable data to investigate REE sorption kinetics on DdTC/PVC sorbent. The two forms of nonlinear equations are offered as follows [41]:(4)$$q_{t}=q_{e}\left(1-e^{k_{1}t}\right)$$
(5)$$q_{t}=\frac{k_{2}q_{2}^{2}t}{1+k_{2}q_{2}t}$$

From the nonlinear regression of first-order kinetics presented in Figure 5b, *q*_1_, *k*_1_, and *R*^2^ were 175 mg/g, 0.026 1/min, and 0.98, respectively; the reduced Chi-square was 75.76. However, *q*_2_, *k*_2_, and *R*^2^ were 240.23 mg/g, 8.82 g/mg.min, and 0.95, respectively; the reduced Chi-square was 113.17 for the nonlinear regression of second-order kinetics. From the data in Figure 5b, *R*^2^ for first-order was precise and close to unity, and *q*_2_ was 175 mg/g, close to the experimental uptake (*q_e_* is 156.56 mg/g). Therefore, it is implied that the first-order kinetic model describes rare-earth ions adsorption kinetics upon DdTC/PVC adsorbent. Therefore, REE adsorption by DdTC/PVC is most likely controlled by the chemisorption process [42].

#### 3.2.4. Initial REE Concentration and Equilibrium Isotherm

Initial REE concentration is the basic parameter in the adsorption approaches that controls the adsorption behaviors of rare-earth ions. Several batch experiments were applied to understand the role of initial concentration upon the sorption uptake. The experiments were performed using 50 mL RE ions in the range of 25–400 mg/L, at pH 5.5, and at room temperature. Figure 6a reveals that when the RE ions’ initial concentration was raised, RE ion uptake (*q_e_*, mg/g) increased and achieved the maximum loading uptake at 200 mg/L of REE. The maximum uptake for DdTC/PVC was 156.6 mg/g, which remained constant after 200 mg/L. It was shown that the working adsorbent reached its maximum loading capacities (saturation capacities) because RE ions mobility was the highest, and the total active sites of DdTC/PVC were filled and blocked with RE ions in solution.

The adsorption data were shown to illuminate how REEs interact with DdTC/PVC. The most commonly used isotherm models for analyzing adsorption balance are the Langmuir and Freundlich isotherm models. Langmuir’s model considers that the homogeneity of the adsorbent surface is suitable for monolayer adsorption through the DdTC/PVC surface due to it having a limited number of related sites. This model assumes reliable adsorption energies over the surface and no migration of REEs in the DdTC/PVC surface plane [43,44,45]. The Freundlich isotherm model designates the RE ion adsorption on the DdTC/PVC surface. It is naturally employed to examine the surface energies and heterogeneity [46,47,48]. The Langmuir nonlinear form is as follows:(6)$$q_{e}=\frac{q_{max}k_{i}C_{e}}{1+k_{i}C_{e}}$$

In the above formula, *C**_e_* (mg/L) represents the equilibrium REE concentration and *q_max_* is maximum adsorbed RE ions.

The empirical nonlinear Freundlich isotherm model based on REE adsorption through the DdTC/PVC heterogeneous surface was set as:(7)qe=kfCe1n
where *k**_f_* and *n* are RE ion uptake and RE ion adsorption intensity. From Figure 6b, the adsorption uptake of the Langmuir model (170.42 mg/g) is near to the experimental uptake (156.6 mg/g), and its *R*^2^ (0.96) is near to unity when compared with *k_f_* (41.64 mg/g) and the *R*^2^ (0.87) of the Freundlich model. Furthermore, the Chi-square of the Langmuir model (117.56) is smaller than the Chi-square of the Freundlich (449.46) model. Hence, the Langmuir model is the best to fit for the experimental data. Accordingly, REE uptake on DdTC/PVC adsorbent is compared with other adsorbent materials in Table 2.

#### 3.2.5. Effect of Temperature and Thermodynamics

The role of temperature was investigated on REE adsorption at 25–60 °C. All experiments were executed using 50 mL solution, with 200 mg/L for REEs and pH of 5.5 (Figure 7a). The REE uptake on DdTC/PVC diminished from 156.56 to 127.40 mg/g when the temperature increased due to the DdTC/PVC bond breakdown; therefore, the active sites decreased. Accordingly, 25 °C is the optimum temperature for RE ion adsorption.

The thermodynamic characteristics were determined to analyze the nature/feasibility of the adsorption method. Thermodynamic factors such as changes in free energy (Δ*G*°), enthalpy (Δ*H*°), and entropy (Δ*S*°) are evaluated in Table 3 and Figure 7b. The equilibrium constant *K_d_* was determined from the experimental data of REE adsorption on DdCT/PVC. These factors were evaluated via these equations [55,56,57,58]:(8)$$logK_{d}=\frac{\Delta S}{2.303R}-\frac{\Delta H}{2.303RT}$$
(9)ΔG=ΔH−TΔS

The thermodynamic parameters of DdCT/PVC were evaluated by the relation of log *K_d_* vs. 1/T. From the data, negative Δ*G*° values indicate that REE adsorption upon DdTC/PVC was achievable and spontaneous. Likewise, free energy interactions demonstrated that adsorption reactions are promising for chemical interactions between REEs and DdTC/PVC adsorbent. Furthermore, the negative Δ*S*° value confirms the adsorption feasibility and randomness at the interface of REE adsorption on the DdCT/PVC surface.

### 3.3. Rare Earth Desorption

REE desorption is a crucial stage in designing a sorption approach; it is definitely the most acceptable way to improve REE uptake. Moreover, it is also vital for the DdTC/PVC recyclability experiment. As in the adsorption process, many factors affected the desorption efficiency in batch techniques, such as the desorbing agent concentration, the S:L phase ratio, and the contact time.

#### 3.3.1. Desorbing Agent Type

The effect of desorbing agents on the desorption of REE ions from REEs/DdCT/PVC was analyzed out by shaking desorbing agents (HNO_3_, HCl, H_2_SO_4,_ and NaCl). The other desorption factors were constant, with 0.5 M desorbing concentration, 1:50 S:L ratio (5 mL desorbing to 0.1 g DdCT/PVC), 30 min contact time, and ambient temperature. From Figure 8a, the REE desorption efficiency using 0.5 M HCl reached maximum desorption (68.3%). Therefore, HCl should be used for the quantitative desorption of REEs.

#### 3.3.2. HCl Concentration

The REE desorption from the REEs/DdCT/PVC was carried out using various HCl concentrations ranging from 0.1 to 2.5 M. The relevant limitations were constant in these experimentations, using 0.1 g DdCT/PV and 5 mL HCl for 30 min at ambient temperature. From Figure 8b, the desorption efficiency of REEs increased with the increasing HCl concentration up to 1.0 M, whereas the maximum desorption attained 87.1%. Consequently, 1.0 M HCl was used for the REE desorption process.

#### 3.3.3. Contact Time

A series of experiments were done to check the influence of contact time on the REE desorption by shaking 0.1 g of REEs/DdTC/PVC with 5.0 mL 1.0 M HCl for different times (30–120 min). Figure 8c shows that 45 min contact time was required for maximum REE desorption (91.2%).

#### 3.3.4. S:L Phase Ratio

To investigate the minimum volume of HCl solution for the REE desorption from the loaded DdTC/PVC, the S:L ratio were analyzed in a range of 1:10 to 1:80. At the same time, the other factors were constant, with 1.0 M HCl for 45 min contact time. Figure 8d found that the rare earth ion desorption efficiency was increased by decreasing the S:L ratio until 1:60; after that, the REE desorption efficiency was almost constant, at 98.0%. Summing up, the optimum conditions for the desorption of REEs from the loaded adsorbent were 1 M HCl, 45 min contact time, and 1:60 S:L ratio.

### 3.4. Regeneration of DdTC/PVC Adsorbent

To reuse and recycle the DdTC/PVC adsorbent, it was regenerated with 1.0 M HCl and a 1:60 S:L ratio for 45 min contact time. After eight consecutive cycles, the adsorption-desorption processes were repeated many times until the desorption was decreased from 98.0 to 80.5% of the DdTC/PVC. It was decided that DdTC/PVC has a good adsorption constancy for REE recovery.

### 3.5. Leaching Studies

#### 3.5.1. Characterization

The lamprophyre dike sample was collected from the Abu Rusheid region. The XRD outline of the lamprophyre dikes is shown in Figure 9a. The acquired results show that lamprophyre dikes are composed of quartz, kaolinite, K-feldspars, plagioclases, biotite, and muscovite. The XRD spectrum after heavy metals separation by density separation using bromoform is shown in Figure 9b. After heavy minerals separation, the mineralizations were composed of REEs and Nb-Ta-bearing minerals (tantalite, xenotime, zircon, fergusonite, monazite, apatite allanite) [59]. Complete chemical analysis was applied on the lamprophyre dike sample to determine major and trace ion concentrations. These results clearly show that SiO_2_, Fe_2_O_3_, Al_2_O_3_, K_2_O, P_2_O_5_, and CaO assayed 46.3, 15.36, 16.8, 2.8, 1.02, and 2.3%, respectively, while the yttrium and uranium contents of this sample assayed 3274 and 400 mg/kg. Furthermore, individual REEs were determined by the ICP-OES technique (Table 4). REE analysis showed that the working sample is rich in heavy REEs over light REEs.

#### 3.5.2. Leaching Investigation

Leaching is an essential process in rare earth ore processing in which the solutes are transferred from a solid to a leach liquor. Previous mineralogical and chemical analysis results of the lamprophyre dyke sample emphasized the presence of valuable elements such as REEs. The sample was first crushed and ground to a definite size to study REE leaching. Leaching experiments were studied at a consistent HCl concentration and solid/liquid ratio for different times and temperatures.

##### HCl Concentration

A series of experiments were executed to find the optimum HCl concentration for REE leaching. The concentration varied from 0.5–5.0 M under agitating speed of 220 rpm, S/L ratio of 1/3, and leaching time of 120 min. The REE leaching efficiency was enhanced from 42.27 to 76.92% by raising the HCl concentration from 0.5 to 3.0 M (Figure 10a). Undesirable impurities were dissolved with increasing HCl concentration from 3.0 to 5.0 M, most significantly iron species.

##### Solid/Liquid Ratio

The influence of the S/L ratio was studied from 1/2 to 1/7 at constant conditions of 3.0 M HCl and 220 rpm agitation rate for 120 min agitating time (Figure 10b). REE leaching efficiency increased from 70.46 to 87.64% by decreasing the S/L ratio to 1/4; after that, the REE leaching efficiency was constant. Consequently, the optimum ratio chosen for the subsequent experiments was 1/4.

##### Agitation Time

This parameter of REE leachability was examined to identify the agitating time required for maximum leaching efficiency of REEs. It was conducted for times ranging from 30–300 min at 3.0 M HCl concentration, 1/4 S/L ratio, and 220 rpm agitation rate. The data in Figure 10c indicates that the REE leaching efficiency reached 92.0% after 180 min of agitating time; after that time, there was no growth in the REE leachability.

##### Leaching Temperature

This factor was examined from 25 to 90 °C at 3.0 M HCl, 220 rpm stirring speed, 1/4 S/L ratio, and 180 min agitating time. The data in Figure 10d reveal that the REE leaching efficiency was slightly improved from 94.2 to 97.4% by raising the leaching temperature from 25 to 90 °C. Thus, 25 °C was chosen for REE leaching to avoid hydrolysis and high operating cost.

### 3.6. Application

According to optimum conditions, the leach liquor of REEs was created by curing 2.0 kg of the suitably pulverized lamprophyre dyke sample with 8.0 L of 3.0 M HCl, with a 180 min agitation time at 25 °C. The acquired leachate was assayed, with 94.2% of REE leaching. The analyses of RE ions and some associated ions in the leachate are presented in Table 5.

#### 3.6.1. Recovery of REEs from Leach Liquor

The chemical analysis of leach liquor showed some impurities, such as Fe^3+^, Al^3+^, Cu^2+^, Pb^2+^, and Zn^2+^, which significantly interfered with the adsorption and precipitation of the REEs. Therefore, these interfering ions should be removed before recovering the REEs. Thus, copper and aluminum hydroxides were separated from the effluent leach liquor by adjusting the pH to 5.5 using ammonia solution. The two hydroxides were separated by adding excess ammonia solution to dissolve selective Cu^2+^ ions. The dissolved copper ions were re-precipitated at pH 5.5 using diluted H_2_SO_4_ [60]. Zinc content was selectively recovered from the leach liquor as Zn sulfide at low pH values [61,62]. According to Veeken et al. [63] and Esposito et al. [64], the precipitation of zinc sulfide could be better realized, with a higher recovery of the metal ions with a lower retention time at low pH, as compared to the metal hydroxide. This was eventually precipitated at pH 2.0 using dropwise 2.0% Na_2_S solution. After filtration, the gray ZnS residue was cleaned several times with water to eliminate any retained impurities. Most iron could be separated as hydroxide by raising the pH to 3.5.

Finally, the lamprophyre dyke leach liquor was introduced to recover REEs using DdTC/PVC adsorbent under the previously found optimum conditions. Under optimum desorption conditions, the loaded adsorbent (REEs/DdTC/PVC) was subjected to desorption of 98% of REEs.

#### 3.6.2. Precipitation of REEs by Oxalic Acid

The precipitation of REEs was performed from the solution after desorption and preconcentration. It used 20.0% oxalic acid to produce rare earth oxalate. For this purpose, after increasing the pH to 3.5, oxalic acid was gradually added with constant stirring at pH 1. Rare earth oxalate was precipitated and then dried. The obtained precipitate was analyzed using ICP-OES (Table 6) and EDX analysis to identify its individual REE distribution (Figure 11).

### 3.7. Group Separation of REEs by Three-Liquid-Phase Extraction

RE ion three-liquid-phase partition behaviors were scrutinized to develop a possible extraction technique to extract rare-earth elements in the groups light, intermediate, and heavy. The RE partition parameters such as pH, polymer amount, complexing agents, and salt concentration on the 3-phase partitioning of three groups of RE ions were scrutinized.

#### 3.7.1. REE Partition Parameters on the Three-Liquid Phase

##### Effect of pH on the Molar Ratio of RE Separation

It is apparent in Figure 12a that when the pH was carefully kept at pH 2.5, almost all light rare-earth ions (LRE) stayed in the bottom phase, which was (NH_4_)_2_SO_4_-rich. Thus, three liquid phase extraction (TLPE) of light rare-earth ions (LRE) from middle and heavy rare-earth ions was achieved. A decrease in the mass fraction of MRE and HRE ions was observed due to these ions’ existence inside the middle phase, which was polyethylene glycol (PEG)-rich, due to the metal ions’ interactions with oxygen atoms of PEG chains [65]. The oxygen atoms were employed as electron donors. The rising pH resulted in the strong hydration of RE ions and prevented the RE ions from reacting with oxygen atoms.

Nevertheless, the added diethylenetriamine penta-acetic acid (DTPA) could be complexed with RE ions, and hence, the hydration of RE ions was stopped. It was supposed that the DTPA interaction with oxygen atoms occurred due to the migration of RE-DTPA complexes inside the PEG-rich intermediate phase [66]. When the pH was more than 2.5, the HRE and MRE ions’ complexation with DTPA was more powerful than with Cyanex 272; however, the LRE ions’ complexation was not improved with DTPA. The LRE ions’ mass fraction rose only at a pH above 3.0, as it did without adding DTPA. DTPA acquisition allowed competitive complexation of DTPA and Cyanex 272 with RE ions. It guided the extraction of HRE ions from MRE and LRE ions.

##### Effect of DTPA on the Molar Ratio of RE Extraction

The separation of RE ions in the PEG middle phase might apply to PEG’s complexing ability with RE ions. The oxygen and nitrogen atoms in DTPA formed a chelating ring using RE ions [59], which improved the RE complexes’ stability. In this way, RE ions were complexed via ligand and were pulled inside the PEG middle phase. Various RE ion separations in the PEG-middle phase were correlated to the RE complexes’ stability constants. MRE and HRE ions were more feasible to complex with DTPA than LRE ions. Accordingly, almost no LRE ion extraction in the PEG middle phase occurred.

The effect of DTPA quantity upon RE ion extraction was investigated from the data shown in Figure 12b. It was seen that MRE and HRE ions ‘mass fractions in the PEG middle phase improved by increasing the DTPA amount. However, when adding more than a 1:1 ratio of DTPA, MRE, and HRE ion mass fractions stopped increasing, while LRE inside the PEG middle phase was not more than 5.0%, regardless of the amount of DTPA. It was realized that RE-DTPA formation promotes RE ion extraction inside the PEG middle phase complexation interaction between DTPA and RE ions using a molar ratio of 1:1. However, when DTPA was insufficient, the complexation interaction was not finished. Thus, the distribution of MRE and HRE ions inside the PEG middle phase was lowered. In the TLPE processes, the following equations clarified the extraction with Cyanex 272 and DTPA complexation by RE ions.
$${\mathrm{RE}}^{3+}+3{\left(H_{2}A_{2}\right)}_{\left(O\right)}⇋{\left[\mathrm{RE}{\left({\mathrm{HA}}_{2}\right)}_{3}\right]}_{\left(O\right)}+3H^{+}\;{\mathrm{RE}}^{3+}+H_{3}D^{2-}⇌\mathrm{RE}.{\mathrm{HD}}^{-}+2H^{+}$$

Most HRE ions were extracted in the top (Cyanex 272) organic phase. With increasing DTPA, the DTPA complexation with Yb, Lu, and Y contended with Yb, Lu, and Y extraction via the Cyanex 272 top phase. Even though most Yb, Lu, and Y were extracted inside the top organic (Cyanex 272) phase, the complexation transfer was partially inside the PEG middle phase.

##### Effect of the Type of Polymers

The RE ions’ extraction behaviors inside the polymer middle phase established on different polymers are depicted in Figure 12c. The data established that MRE and HRE ions’ mass fractions inside the polymer-middle phase were diminished by increasing the molecular weight of polymer (PEG). At 600 PEG molecular weight, MRE ion extraction was more than 50%. However, at 6000 PEG molecular weight, MRE and HRE ions’ mass fractions declined to 23 and 10%. At the same time, most LRE ions stayed in the salt-bottom water phase, and their extraction in the PEG middle phase did not exceed 10%.

The PEG molecular weight increase could exhaust its terminal OH groups and enrich the interaction among PEG molecules. Thus, the lowest PEG molecular weight offered more strong hydrophilicity [67]. The PEG middle phase hydrophilicity established upon PEG 600 was more powerful, which promoted MRE ion separation.

##### (NH_4_)_2_SO_4_ Content Impact

The effect of amount of (NH_4_)_2_SO_4_ on the RE ion extraction inside the bottom salt and PEG phases is presented in Figure 12d,e. The obtained data establishes that the RE ions’ extraction inside the bottom salt layer improved with increasing salt concentration. The data also illustrates that the LRE ion extraction improved by increasing the (NH_4_)_2_SO_4_ concentration to 1.0 M. Above this concentration, (NH_4_)_2_SO_4_ did not influence LRE extraction in the bottom phase. However, increasing the salt concentration in the aqueous layer crushed the PEG molecules’ hydrated layer. The water molecules were transferred from the PEG middle layer to the salt bottom layer by lowering PEG hydrophilicity [68]. Thus, this influenced the extraction of hydrophilic RE-DTPA complexes. In TLPE, MRE ion extraction inside the PEG middle diminished with increasing salt concentration. The HRE ion extraction inside the salt bottom layer also increased with increasing salt concentration. This might be owing to the enriched HRE ions’ interaction with the redundant amount of PEG in the aqueous solution.

Experimental results indicated that heavy, middle, and light RE ions have distinct extraction behaviors in the top organic, middle polymer, and bottom salt layers of TLPE. At aqueous pH 2.5, DTPA contended with Cyanex 272 to form a complex with RE ions. Most HRE ions were extracted to the top Cyanex 272 layer; MRE ions were extracted to the PEG middle layer, but the maximum LRE ions stayed in the salt bottom layer. The addition of DTPA might improve the MRE ion distribution ratio inside the PEG layer, but the MRE ions’ mass fraction did not improve even after more addition of DTPA. The MRE ions’ mass fraction improved with diminishing polymer molecular weight and increased with increasing quantities of polymer. Increasing the (NH_4_)_2_SO_4_ concentration diminished the MRE ions’ extraction inside the PEG phase. Thus, it is possible to present a one-stage extraction of LRE, MRE, and HRE ions by TLPE. The current guideline implied that RE ion extraction relied upon aqueous pH. Thus, RE ion stripping from the loaded top Cyanex 272 and middle polymer layers in TLPE could be attained.

The following procedures were followed to strip the MRE ions from the PEG middle layer. Hydrochloric acid was put in contact with the RE-loaded PEG layer; this may be popular for the dissociation of RE complexes. Likewise, an adequate amount of (NH_4_)_2_SO_4_ was introduced to the aqueous layer to construct two PEG and (NH_4_)_2_SO_4_ layers, therefore allowing the PEG to be reused and regenerated. Moreover, HRE ion stripping from the loaded organic (Cyanex 272) layer was recorded with 3.0 M HCl [69]. The LRE ions in the salt bottom layer could be recovered with the conventional precipitation method. After removing LRE ions, the salt raffinate could be reused for subsequent TLPE.

#### 3.7.2. Light, Middle, Heavy RE Ion Recovery

Finally, LRE, MRE, and HRE ion separation was performed by TLPS (Cyanex 272/PEG_600_/(NH_4_)_2_SO_4_−H_2_O) from the solution of the prepared rare earth oxalate concentrate after dissolving it in diluted HCl. Hence, separating RE ions into three groups is possible.

The practical outcomes implied that HRE ions (Yb, Lu, and Y) would selectively extract inside the top organic layer. MRE ions (Gd, Ho, and Er) would also extract inside the PEG middle layer. Extraction of LRE ions (La, Ce, Nd, and Sm) could be enhanced in the (NH_4_)_2_SO_4_ bottom aqueous layer. The middle and heavy rare earth ions were stripped with 3 M HCl and then precipitated with oxalic acid. Additionally, the precipitate of each group was calcined at 700 °C to obtain CeO_2_, La_2_O_3_, Nd_2_O_3_, Sm_2_O_3_, Gd_2_O_3_, Ho_2_O_3_, Er_2_O_3_, Yb_2_O_3_, Lu_2_O_3_, and Y_2_O_3_. The light, middle, and heavy rare earth oxides were analyzed using EXD semi-quantitative analysis (Figure 13), and the ICP-OES technique confirmed their constituents. Therefore, by ICP-OES analysis, the light rare earth oxides group was La_2_O_3_, CeO_2_, Nd_2_O_3_, and Sm_2_O_3_, assaying 45.5, 31.7, 11.4, and 8.3%, respectively, while the middle rare earth oxides group was Gd_2_O_3_, Ho_2_O_3_, and Er_2_O_3_, assaying 15.4, 65.5, and 7.5%, respectively. The heavy rare earth oxides group was Yb_2_O_3_, Lu_2_O_3_, and Y_2_O_3_, assaying 10.4, 40.6, and 43.8%, respectively. 

## 4. Conclusions

The present experimental investigations were established to extract RE ions in three groups (LRE, MRE, and HRE ions) by TLPE from rare earth oxalate concentrate obtained from the extraction of RE ions from Lamprophyre dyke leachate using DdTC/PVC sorbent. The latter was prepared by sodium diethyldithiocarbamate and polyvinyl chloride, and it was used for RE ion adsorption. The optimum sorption parameters were pH 5.5, 200 mg/L REEs, 50 mg DdTC/PVC dose, and 60 min sorption time. The maximum uptake was obtained at 156.50 mg/g. Furthermore, kinetic validations were used to fit the results into the first-order nonlinear model. The Langmuir model was also suited for adsorption processes. Thermodynamic changes were explored, showing the negative ∆S° and negative ∆H°, which demonstrated RE ions’ adsorption randomness and exothermic predictability; moreover, the negative ∆G° showed that the adsorption processes were spontaneous. The sorption-desorption cycles were repeated frequently until desorption reduced to 80.0% at eight cycles. Lastly, the LRE, MRE, and HRE ions separated by TLPS were applied to the obtained rare earth oxalate concentrate solution after dissolving in diluted HCl. Finally, this study introduced a new technique that is easy to apply in extracting REEs from their resources, with a good result for RE ion separation in three groups with high purity as rare earth oxides (98%).

## Figures and Tables

**Figure 1 materials-15-01211-f001:**
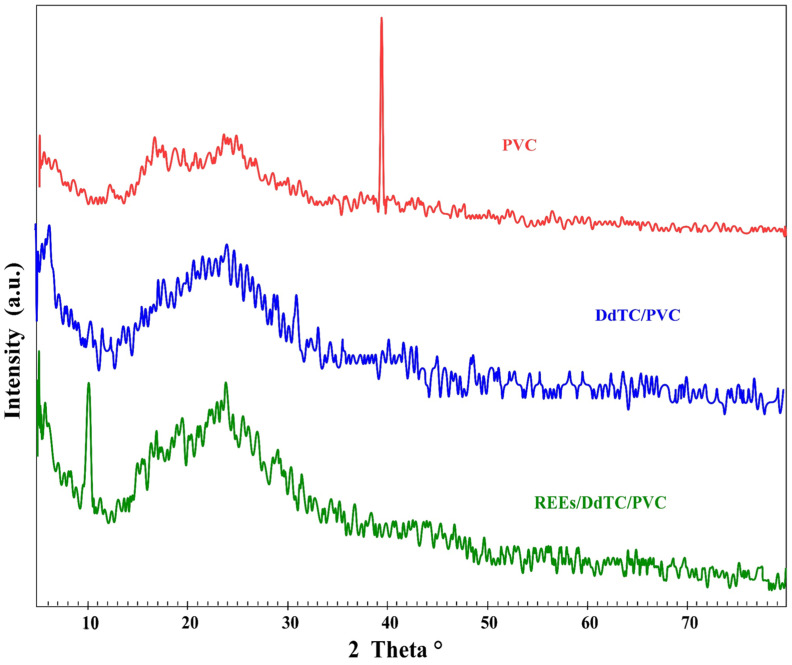
XRD of PVC polymer, DdTC/PVC, and REEs/DdTC/PVC.

**Figure 2 materials-15-01211-f002:**
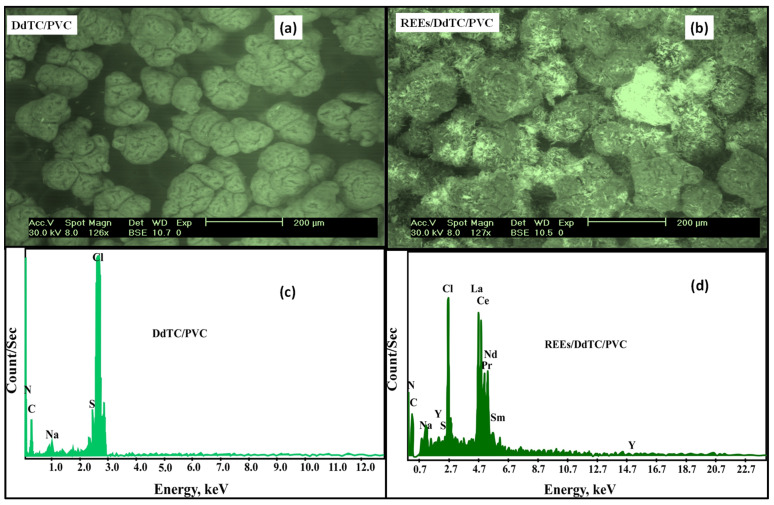
(**a**,**b**) SEM and (**c**,**d**) EDX of DdTC/PVC and REEs/DdTC/PVC.

**Figure 3 materials-15-01211-f003:**
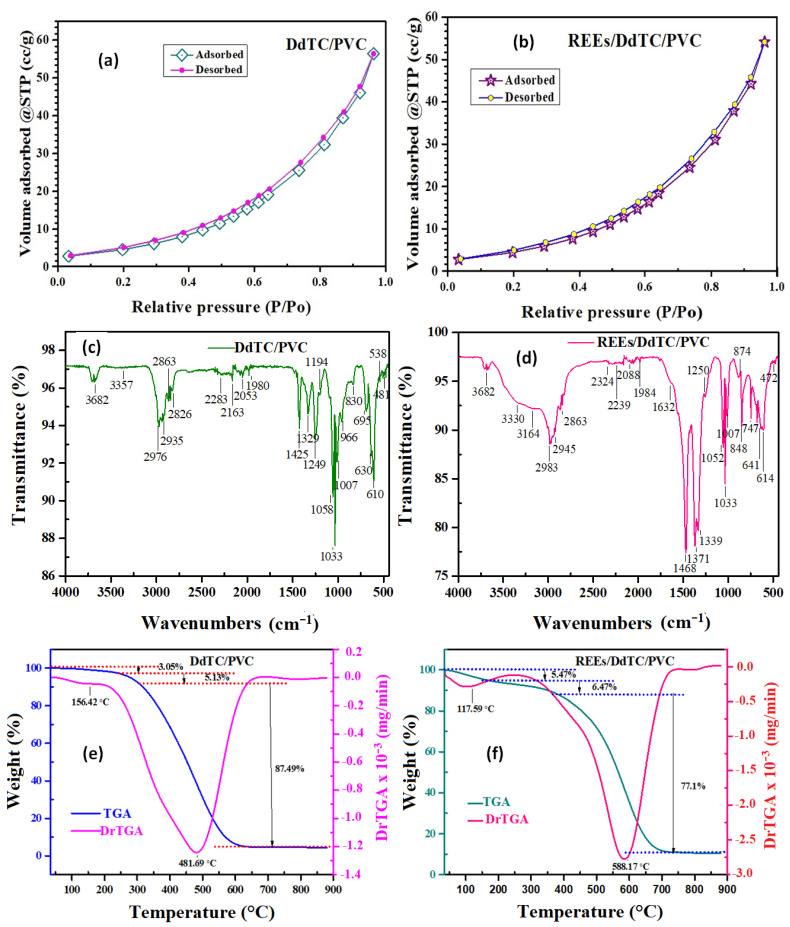
(**a**,**b**) N_2_ sorption/desorption isotherms, (**c,d**) FTIR spectra, (**e**,**f**) TGA of DdTC/PVC and REEs/DdTC/PVC.

**Figure 4 materials-15-01211-f004:**
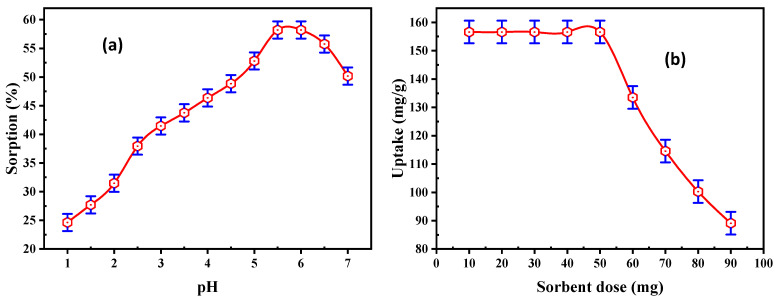
(**a**) Effect of pH and (**b**) effect of adsorbent dose on REE adsorption.

**Figure 5 materials-15-01211-f005:**
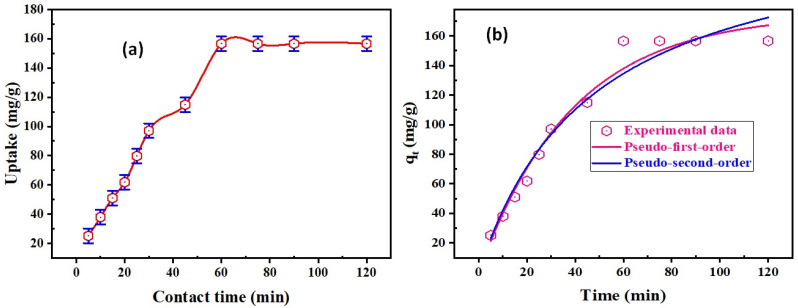
(**a**) Effect of adsorption time upon REE uptake; (**b**) pseudo-first-order and pseudo-second-order nonlinear models for REE uptake on DdTC/PVC.

**Figure 6 materials-15-01211-f006:**
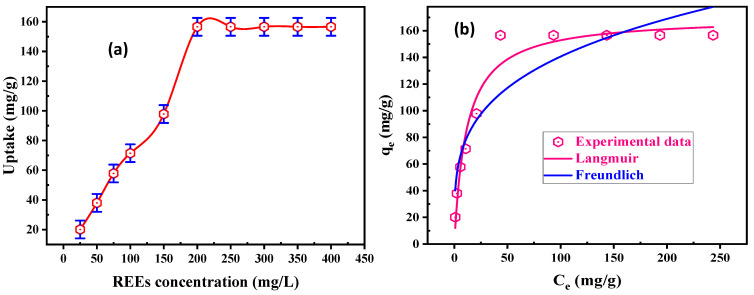
(**a**) Effect of REE concentration on uptake of DdTC/PVC; (**b**) Langmuir and Freundlich nonlinear models for REE adsorption.

**Figure 7 materials-15-01211-f007:**
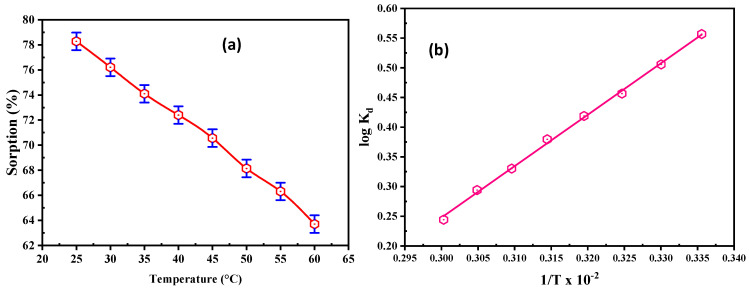
(**a**) Influence of temperature on REE adsorption; (**b**) log *K_d_* vs. 1/T relation for REE adsorption on the DdCT/PVC.

**Figure 8 materials-15-01211-f008:**
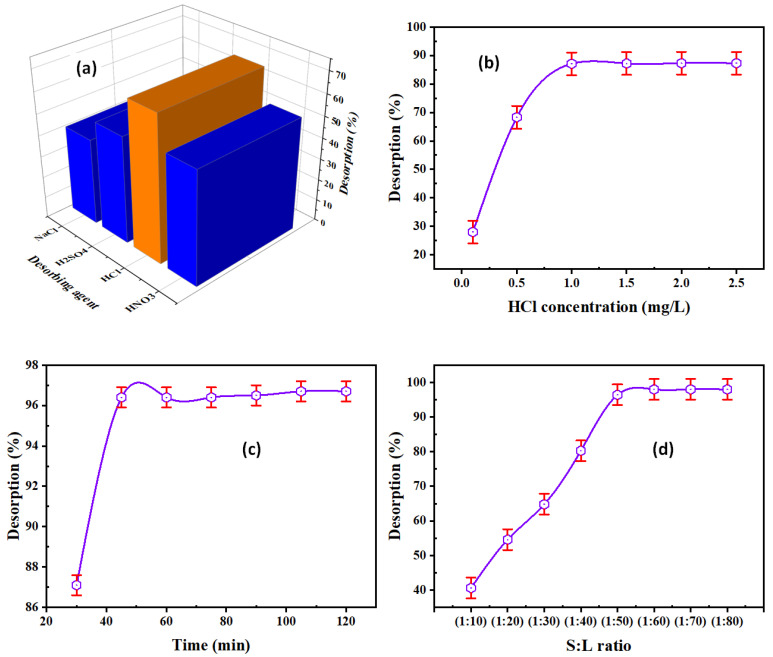
(**a**) Desorbing agent, (**b**) HCl concentration, (**c**) contact time, and (**d**) S:L ratio influence on REE desorption efficiency.

**Figure 9 materials-15-01211-f009:**
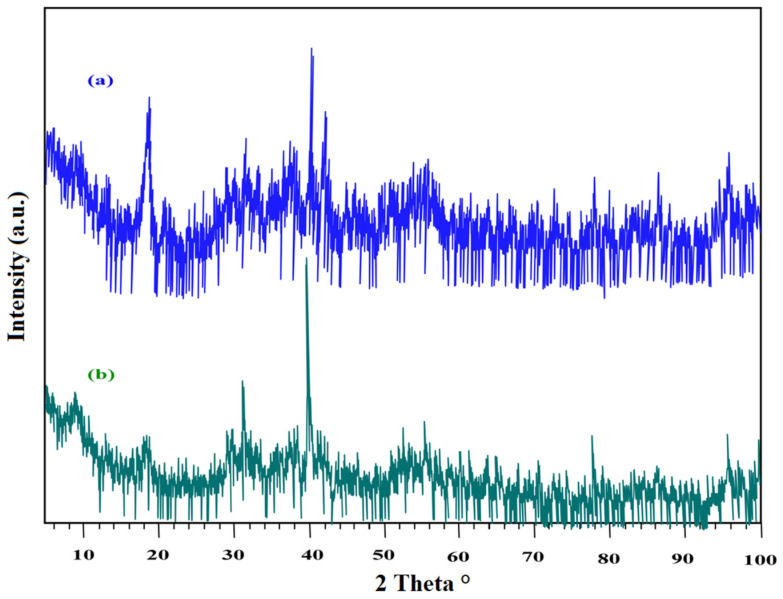
XRD patterns of (**a**) lamprophyre dykes and (**b**) heavy minerals of lamprophyre dykes after density separation.

**Figure 10 materials-15-01211-f010:**
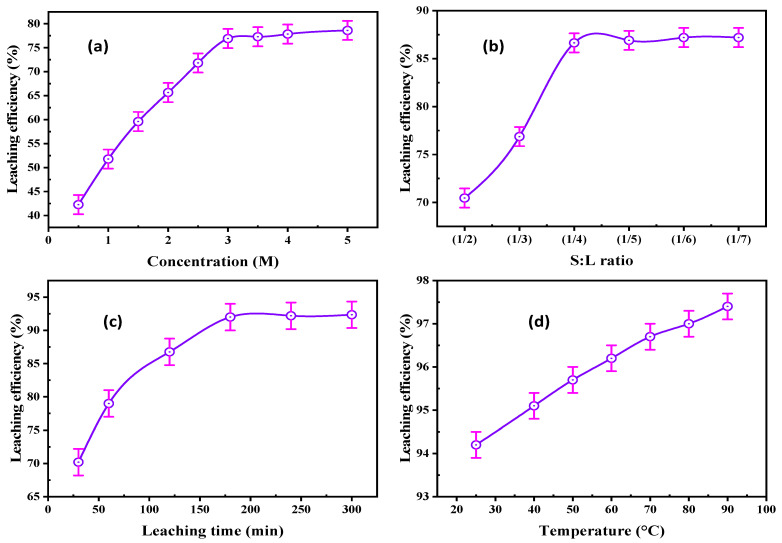
(**a**) Effect of HCl concentration, (**b**) S/L ratio, (**c**) agitation time, and (**d**) leaching temperature upon REE leaching.

**Figure 11 materials-15-01211-f011:**
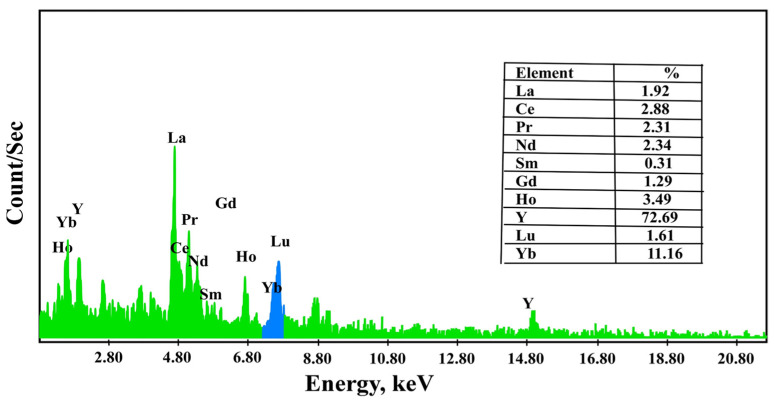
EDX spectrum of rare earth oxalate from REEs loaded onto DdTC/PVC.

**Figure 12 materials-15-01211-f012:**
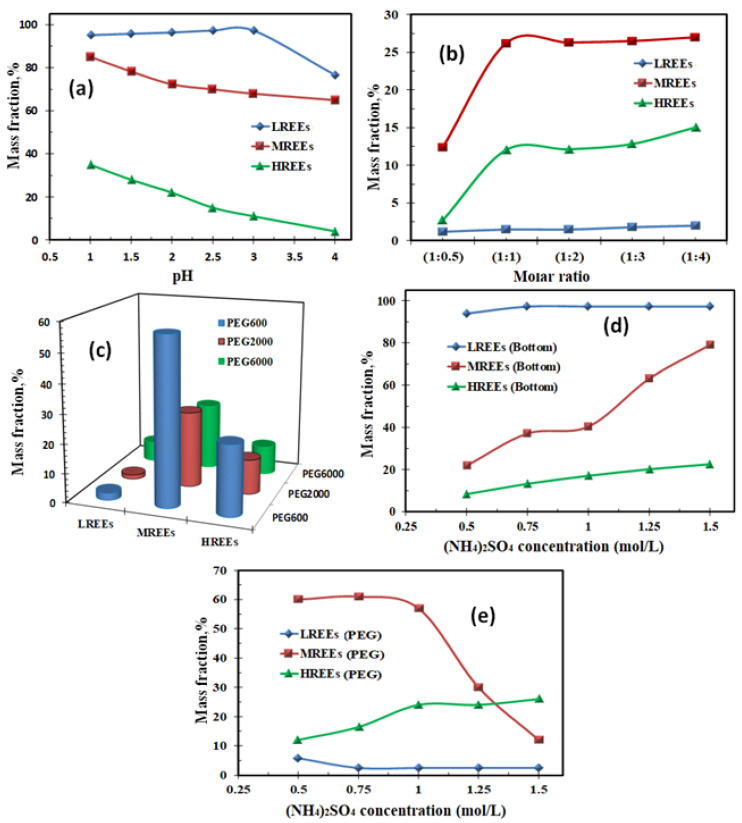
(**a**) Effect of pH on the RE ion extraction inside the bottom layer; (**b**) effect of DTPA molar ratio on the RE ion extraction inside the PEG middle layer; (**c**) effect of polymers type on the RE ion extraction inside the polymer middle layer; (**d**) effect of (NH_4_)_2_SO_4_ concentration on the RE ion extraction inside the bottom salt layer; (**e**) effect of (NH_4_)_2_SO_4_ concentration on the RE ion extraction inside the PEG layer.

**Figure 13 materials-15-01211-f013:**
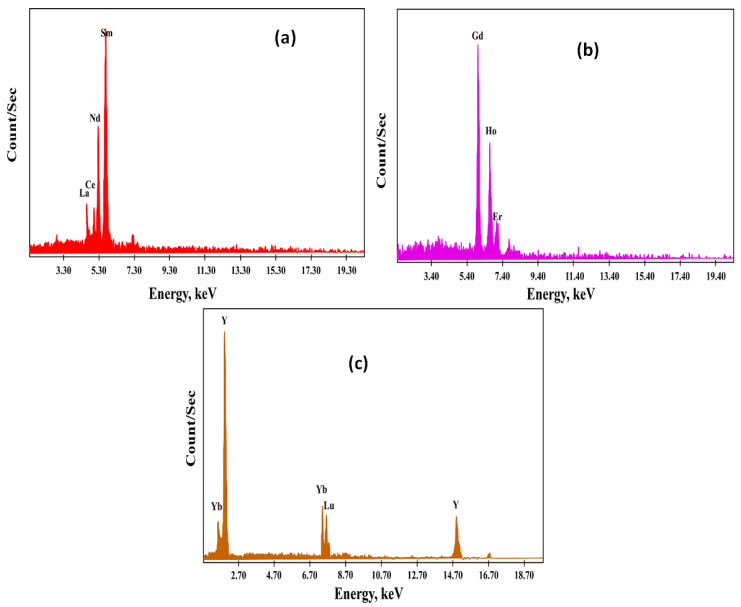
EDX patterns of (**a**) light REEs as a group, (**b**) middle REEs as a group, (**c**) heavy REEs as a group.

**Table 1 materials-15-01211-t001:** Surface area, pore volume, and pore size of DdTC/PVC before and after rare-earth ion adsorption.

Materials	S_BET_, m^2^/g	Pore-Volume, cc/g	Pore-Size, nm
DdTC/PVC	90.14	0.087	1.97
REEs/DdTC/PVC	86.53	0.084	1.89

**Table 2 materials-15-01211-t002:** Comparison of REE uptake of some adsorbent materials and DdCT/PVC adsorbent.

Adsorbent Materials	Uptake, mg/g	Reference
Modified coal fly ash-based SBA-15	32.0	[49]
Poly (vinyl alcohol-co-ethylene) (EVOH) modified polymer	22.05	[50]
Powdered spirulina	72.46	[51]
Magnetic nanocomposite of calcium alginate carrying poly(pyrimidine-thiophene-amide)	113.08	[52]
Poly(amidoxime-hydroxamic acid) resin	125.0	[53]
Graphene oxide-tris(4-aminophenyl)amine composite	30.88	[54]
Sodium diethyldithiocarbamate/polyvinyl chloride (DdTC/PVC)	156.6	This study

**Table 3 materials-15-01211-t003:** Thermodynamic REE adsorption parameters on DdCT/PVC adsorbent.

Parameters	Temperature, K	DdTC/PVC
Δ*G*°, kJ/mol	298	−3.158
	303	−2.938
	308	−2.718
	313	−2.498
	318	−2.278
	323	−2.058
	328	−1.838
	333	−1.618
Δ*H*°, kJ/mol		−16.27
Δ*S*°, kJ/mol K		−0.0440

**Table 4 materials-15-01211-t004:** Chemical analysis of lamprophyre dikes.

Major Oxides	Wt., %	Trace Metal Ions	mg/kg	REE Ions	mg/kg
SiO_2_	46.3	U^6+^	400	La^3+^	90
Al_2_O_3_	16.8	Ba^2+^	167	Ce^3+^	160
TiO_2_	3.4	Pb^2+^	596	Pr^3+^	170
Fe_2_O_3_	15.36	V^5+^	240	Nd^3+^	143
MnO	0.65	Cu^2+^	367	Sm^3+^	26
MgO	0.06	Ni^2+^	86.7	Gd^3+^	60
CaO	2.3	Cd^2+^	49.5	Ho^3+^	171
K_2_O	2.8	Zn^2+^	6348	Er^3+^	145
Na_2_O	0.48	Th^4+^	39	Yb^3+^	550
P_2_O_5_	1.02			Lu^3+^	65
LOI *	8.75			Y^3+^	3274
Total	97.92				

LOI *: Loss of ignition (1000 °C).

**Table 5 materials-15-01211-t005:** Chemical analysis of RE and some metal ions in the leach liquor.

Metal Ions	g/L	Metal Ions	mg/L	RE Ions	mg/L
Si^4+^	1.35	U^6+^	94	La^3+^	21
Al^3+^	2.34	Ba^2+^	44	Ce^3+^	37
Ti^4+^	0.32	Pb^2+^	103	Pr^3+^	40
Fe^3+^	3.57	V^5+^	51	Nd^3+^	34
Mn^2+^	0.5	Cu^2+^	85	Sm^3+^	6
Mg^2+^	0.03	Ni^2+^	19	Gd^3+^	14
Ca^2+^	0.67	Th^4+^	5	Ho^3+^	40
K^+^	0.73	Zn^2+^	620	Er^3+^	34
Na^+^	0.1			Yb^3+^	129
P^5+^	0.12			Lu^3+^	16
				Y^3+^	769

**Table 6 materials-15-01211-t006:** Chemical analysis of RE oxalate products obtained from REEs loaded onto DdTC/PVC adsorbent using ICP-OES.

Metal Ions	%	Metal Ions	%
La^3+^	0.72	Ho^3+^	1.38
Ce^3+^	1.39	Yb^3+^	4.57
Pr^3+^	1.30	Lu^3+^	0.57
Nd^3+^	1.19	Y^3+^	27.89
Sm^3+^	0.21	H_2_O (at 110 °C)	13.52
Gd^3+^	0.51	CO_2_ (at 550 °C)	43.61
Total	96.87		

## Data Availability

Not applicable.

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
