# Peer review of "Rare Earth Group Separation after Extraction Using Sodium Diethyldithiocarbamate/Polyvinyl Chloride from Lamprophyre Dykes Leachate"

_materials, 2022, doi:10.3390/ma15031211_

Round 1
Reviewer 1 Report
This paper studied the rare earth group separation after extraction using sodium diethyldithiocarbamate/polyvinyl chloride from Lamprophyre dykes leachate. It was described by XRD, SEM, TGA, and FTIR. Also, the varied factors pH, initial rare earth concentration, time, and DdTC/PVC dose were entertained. However, the reviewer suggests that some revisions should be done before considering acceptance.
- The abstract should be rewritten, i.e., it should highlight the originality, innovation and uniqueness of this paper.
- The English must be improved substantially.
- The research background is not well summarized, so the “Introduction” of this manuscript should be reorganized. Flotation is a useful method to treat rare earth minerals, so it should be described in the “Introduction”, and several relevant references may be added to support this point, such as Transactions of Nonferrous Metals Society of China 31 (2021) 3564−3578; International Journal of Mining Science and Technology 31 (2021) 1117–1128.
- More details about the experimental conditions should be provided.
- The data in the manuscript should have error analysis.
- It is suggested to compare the results of the present research with some similar studies which is done before.
7. Please clearly indicate what is novel about this paper as compared to many other similar papers in the literature.
Author Response
Dear Reviewer,
Please find attached the submission of the carefully revised version of the manuscript in Ref., following the minor comments and modification of the Reviewer.
Below is a detailed list of the changes made in response to the Reviewer’s minor comments (in italics), which outlines every change made a point by point. The changes are marked in the manuscript text (blue font).
Comments
- The abstract should be rewritten, i.e., it should highlight the originality, innovation and uniqueness of this paper.
The abstract has been rewritten, and we showed it to highlight originality, innovation, and uniqueness.
- The English must be improved substantially.
The English language was improved in the whole manuscript.
- The research background is not well summarized, so the “Introduction” of this manuscript should be reorganized. Flotation is a useful method to treat rare earth minerals, so it should be described in the “Introduction”, and several relevant references may be added to support this point, such as Transactions of Nonferrous Metals Society of China 31 (2021) 3564−3578; International Journal of Mining Science and Technology 31 (2021) 1117–1128.
The research background was re-summarized, and also the flotation technique was added as the following:
Recently, foam flotation methods are a promising naive method of recovering rare earth elements from leachates of primary and secondary resources. Foam flotation of La3+, Ce3+, Gd3+, and Yb3+ ions individually, as a group, and as a group with gangue ions (Al3+, Zn2+, Ca2+, and Mg2+). s. The anionic surfactants sodium dodecyl sulfate (SDS) at basic pH and mono-rhamnolipid at pH9 were used for separating REEs in the presence of gangue cations at a surfactant: total cation (excluding Na+) ratio as low as 1:13 [19]. Rare earth minerals enriched pre-concentrate (after lab-scale gravity and magnetic separation steps) were separated by hydroxamic acid flotation [20]. Besides, Copper oxide sulfidization flotation was applied after modification of cuprite by hydrogen peroxide (as an oxidant to improve its consequent sulfidization [21]. Moreover, the direct sulfidization of cuprite is ineffective because cuprite is a copper-oxide mineral with strong surface hydrophilicity. Hence, sodium hypochlorite as an oxidant was used to modify cuprite surfaces to regulate the sulfidization of cuprite. The flotation recovery of pre-oxidized cuprite was nearly 25% higher than that of direct sulfidization flotation, which indicates that the cuprite surface activity was enhanced after pre-oxidation by Cu(I) species (weak affinity with sulfur ions) transformation to Cu(II) species (strong affinity with sulfur ions) [22].
- More details about the experimental conditions should be provided.
The experimental section was rewritten to introduce more explanation with highlighted a blue color.
- The data in the manuscript should have error analysis.
Error analysis was done in Fig. 4a,b, Fig. 5a, Fig, 6a, Fig. 7a, Fig. 8b,c,d, Fig. 10a,b,c,d.
- It is suggested to compare the results of the present research with some similar studies which is done before.
It was done at the end of page 11, Table 2.
Accordingly, the REEs uptake on DdTC/PVC adsorbent is more convenient when compared with other adsorbent materials, as shown in Table 2.
Table 2. Comparison of REEs uptake of some adsorbent materials and DdCT/ PVC adsorbent.
Adsorbent materials |
Uptake, mg/g |
Reference |
Modified coal fly ash based SBA-15 |
32 |
[49] |
Poly (vinyl alcohol-co-ethylene) (EVOH) modified polymer |
22.05 |
[50] |
Powdered spirulina |
72.46 |
[51] |
Magnetic nanocomposite of calcium alginate carrying poly(pyrimidine-thiophene-amide) |
113.08 |
[52] |
Poly(amidoxime-hydroxamic acid) resin |
125 |
[53] |
Graphene oxide-tris(4-aminophenyl)amine composite |
30.88 |
[54] |
Sodium diethyldithiocarbamate/polyvinyl chloride (DdTC/PVC) |
156.6 |
This study |
- Please clearly indicate what is novel about this paper as compared to many other similar papers in the literature.
This study introduced a separation of rare earth elements for leach liquor that gained from the rock sample as three groups (light, middle, and heavy rare earth elements) in pure forms according to their constituents in the original rock sample. The group separation was performed after REEs adsorption and desorption on sodium diethyldithiocarbamate/polyvinyl chloride (DdTC/PVC) as a novel adsorbent from Lamprophyre dykes leach liquor. This technique is easy to use and gives a good result in high purity of rare earth separation; it was compared with other adsorbent materials in the previous works.
We thank the Reviewer a lot for the useful and valuable comments that have helped improve the manuscript.
Hoping that all the careful review is sufficient for the direct acceptance of the manuscript, thank you for your time and consideration.
Best wishes,
Reviewer 2 Report
The article entitled "Rare earth group separation after extraction using sodium diethyldithiocarbamate/polyvinyl chloride from Lamprophyre dykes leachate" concerns the application of a new solution/procedure for REE separation based on the use of sodium diethyldithiocarbamate impregnated on polyvinyl chloride (sorbent). The experimental part of the work is extensive. The Authors examined the influence of various factors, e.g. pH, initial rare earth concentration, time, and DdTC/PVC dose, on the effectiveness of the process. They also conducted experiments of the rare-earths desorption from loaded DdTC/PVC by 1 M HCl, and extraction/separation of RE ions in three groups via TLPE.
In my opinion, the article will benefit if the authors introduce the following changes:
- In the "Introduction" section on REE sorption, the Authors refer to various studies based on the use of various chemical compounds, but they do not provide information on the effectiveness of these solutions - it is worth supplementing this information.
- It is worth explaining/emphasizing what is the advantage of the solution used by the Authors in relation to other solutions described in the literature.
- It is necessary to correct many typos in the text, such as unjustified use of capital letters in the middle of a sentence (e.g. chapter 3.1.2 "It can be concluded That the REEs peaks were perceived ..... ", chapter 3.6.1" Besides, Individual REEs were determined ... "and many others), no capital letters at the beginning of the sentence (e.g. page 10, second line" Pseudo-1st- and 2nd-order were sustained for the reliable data to investigate REEs sorption kinetics upon DdTC / PVC sorbent. the two ... "), errors in formulas (e.g. chapter 3.7.1 "as ... Cu2 +, ...", p. 20 "Likewise, an adequate quantity of (NH4)2SO4 was introduced to the aqueous layer to construct two PEG and (NH4)2SO4 ... ").
- In Chapter 3.8.1.1 the abbreviations LRE, MRE, HRE have been used - they should be explained.
- The "Conclusion" section is purely descriptive. It is worth emphasizing what is the novelty of the applied solution, and what impact the results of the research may have on the development of the field.
Author Response
09January 2022
Dear Reviewer,
Please find attached the submission of the carefully revised version of the manuscript in Ref., following the minor comments and modification of the Reviewer.
Below is a detailed list of the changes made in response to the Reviewer’s minor comments, which outlines every change made point by point. The changes are marked in the manuscript text (green font).
Comments
1- In the "Introduction" section on REE sorption, the Authors refer to various studies based on the use of various chemical compounds, but they do not provide information on the effectiveness of these solutions - it is worth supplementing this information.
Different pieces of information on REEs sorption with various extractants were introduced and explained in the different studies with highlighted green color in the Introduction section.
2- It is worth explaining/emphasizing what is the advantage of the solution used by the Authors in relation to other solutions described in the literature.
It was done at the end of page 11, Table 2.
3- It is necessary to correct many typos in the text, such as unjustified use of capital letters in the middle of a sentence (e.g. chapter 3.1.2 "It can be concluded That the REEs peaks were perceived ..... ", chapter 3.6.1" Besides, Individual REEs were determined ... "and many others), no capital letters at the beginning of the sentence (e.g. page 10, second line" Pseudo-1st- and 2nd-order were sustained for the reliable data to investigate REEs sorption kinetics upon DdTC / PVC sorbent. the two ... "), errors in formulas (e.g. chapter 3.7.1 "as ... Cu2 +, ...", p. 20 "Likewise, an adequate quantity of (NH4)2SO4 was introduced to the aqueous layer to construct two PEG and (NH4)2SO4 ... ").
It was done
4- In Chapter 3.8.1.1 the abbreviations LRE, MRE, HRE have been used - they should be explained.
It was done on page 20 as the following:
It could be apparent in Figure 12a. If the pH was carefully kept at pH2.5, almost all light rare-earth ions (LRE) stayed in the bottom phase that riched with (NH4)2SO4. Thus, three liquid phase extraction (TLPE) of light rare-earth ions (LRE) from middle rare-earth ions (MRE) and heavy rare-earth ions (HRE) were achieved. The decrease in mass fraction of MRE and HRE ions was observed due to these ions' existence inside the middle phase that riched with polyethylene glycol (PEG) due to the metal ions' interactions with oxygen atoms of PEG chains
5- The "Conclusion" section is purely descriptive. It is worth emphasizing what is the novelty of the applied solution, and what impact the results of the research may have on the development of the field.
The conclusion section was rewritten as the following:
The present experimental investigations were established to extract RE ions in three groups (LRE, MRE, and HRE ions) by TLPE from rare earth oxalate concentrate gained from the extraction of RE ions from Lamprophyre dykes leachate using DdTC/PVC sorbent. The latter was prepared by sodium diethyldithiocarbamate and polyvinyl chloride, and it was operated to RE ions adsorption. The optimum sorption parameters were pH5.5, 200 mg/L REEs, 50 mg DdTC/PVC dose, and 60 minutes sorption time. The maximum uptake was gained at 156.50 mg/g. Further, kinetic validations were specified to outfit into the first-order nonlinear model. Also, the Langmuir model was suited for adsorption processes. Also, thermodynamically changes were explored the negative ∆S° and negative ∆H° that recognized RE ions adsorption randomness and exothermic predictable; moreover, negative ∆G° cleared that the adsorption processes were spontaneous. The sorption-desorption cycles were frequently repeated until desorption was reduced to 80.0% at eight cycles. Lastly, the LRE, MRE, and HRE ions separated by TLPS were applied to the gained rare earth oxalate concentrate solution after dissolving in dilute HCl. Finally, this study introduced a new technique that is easy to apply in REEs extraction from their resources with a good result for RE ions separation in three groups with high purity as rare earth oxides (98%).
We thank the Reviewer a lot for the useful and valuable comments that have helped improve the manuscript.
Hoping that all the careful review is sufficient for the direct acceptance of the manuscript, thank you for your time and consideration.
Best wishes,